# Effects and Impacts of Different Oxidative Digestion Treatments on Virgin and Aged Microplastic Particles

**DOI:** 10.3390/polym14101958

**Published:** 2022-05-11

**Authors:** Ilaria Savino, Claudia Campanale, Pasquale Trotti, Carmine Massarelli, Giuseppe Corriero, Vito Felice Uricchio

**Affiliations:** 1Italian National Council of Research, Water Research Institute, 70132 Bari, Italy; ilaria.savino@ba.irsa.cnr.it (I.S.); carmine.massarelli@ba.irsa.cnr.it (C.M.); vito.uricchio@ba.irsa.cnr.it (V.F.U.); 2Department of Biology, University of Bari Aldo Moro, 70121 Bari, Italy; giuseppe.corriero@uniba.it; 3Sezione di Entomologia e Zoologia Agraria, Dipartimento di Scienze del Suolo, della Pianta e degli Alimenti, University of Bari Aldo Moro, 70121 Bari, Italy; pasquale.trotti@uniba.it

**Keywords:** microplastics, oxidative digestion, Fenton’s reagent, virgin, aged, weathering, SEM, FTIR

## Abstract

Although several sample preparation methods for analyzing microplastics (MPs) in environmental matrices have been implemented in recent years, important uncertainties and criticalities in the approaches adopted still persist. Preliminary purification of samples, based on oxidative digestion, is an important phase to isolate microplastics from the environmental matrix; it should guarantee both efficacy and minimal damage to the particles. In this context, our study aims to evaluate Fenton’s reaction digestion pre-treatment used to isolate and extract microplastics from environmental matrices. We evaluated the particle recovery efficiency and the impact of the oxidation method on the integrity of the MPs subjected to digestion considering different particles’ polymeric composition, size, and morphology. For this purpose, two laboratory experiments were set up: the first one to evaluate the efficacy of various digestion protocols in the MPs extraction from a complex matrix, and the second one to assess the possible harm of different treatments, differing in temperatures and volume reagents used, on virgin and aged MPs. Morphological, physicochemical, and dimensional changes were verified by Scanning Electron Microscope (SEM) and Fourier Transformed Infrared (FTIR) spectroscopy. The findings of the first experiment showed the greatest difference in recovery rates especially for polyvinyl chloride and polyethylene terephthalate particles, indicating the role of temperature and the kind of polymer as the major factors influencing MPs extraction. In the second experiment, the SEM analysis revealed morphological and particle size alterations of various entities, in particular for the particles treated at 75 °C and with major evident alterations of aged MPs to virgin ones. In conclusion, this study highlights how several factors, including temperature and polymer, influence the integrity of the particles altering the quality of the final data.

## 1. Introduction

Microplastics (MPs) are “synthetic solid particle or polymeric matrix, with regular or irregular shape from 1 µm to 5 mm size, of either primary or secondary manufacturing origin” [1]. Their presence has been reported in all environmental matrices, becoming an emerging problem worldwide [2,3,4]. Due to their small size, high volume surface ratio, and their ability to adsorb or release pollutants [5], MPs’ threats mainly concern their effects on organisms and human health [6,7,8,9]. Therefore, MPs monitoring is important to understand their presence in the environment. Microplastic studies require several methodological approaches to isolate, identify and quantify particles spread in environmental matrices [10,11,12].

However, when environmental matrices are rich in organic matter, a chemical digestion treatment is necessary to remove it and release particles. Organic residues have a density similar to that of polymers, and they may float together MPs during the density separation phase, hindering extraction and quantitative analysis of particles [13]. Organic digestion treatments may be based on oxidizing agents, acids, basics, or enzymes [10,14,15,16]. However, not all procedures remove organic matter without damaging polymers [17,18,19].

Applying strong acids, such as nitric acid (HNO_3_), produced efficient digestion of biota but they are toxic, corrosive, and cause polymers degradation such as polystyrene, polyamide, and polyethylene [10,20,21]. Alternatively, studies have used alkaline solutions and enzymes for biota digestion, but these require much time, may damage some polymers, and are very expensive [22,23,24].

Oxidizing agents are increasingly used for water, soil, and sediment because the type of organic matter is more difficult to digest (leaves, woody debris, algae, etc.) [13,25,26]. However, at high concentrations and temperatures, agents such as hydrogen peroxide (H_2_O_2_) could destroy polyamide particles, reduce their size and alter the colour of polypropylene particles [27]. Digestion protocols should have minimal impact on the morphology, colour, and weight of MPs [25,28].

Several studies analyzed the effects of digestion treatments on MPs, testing different reaction times, temperatures, and regent volumes [19,27,29,30,31,32,33]. However, most methodological studies tested treatments on virgin MP rather than aged, neglecting their effect on fragile and damaged particles, more representative of reality [28,29,30]. Despite the recent development of biodegradable plastics, less impactful on the environment [34], many biotic and abiotic factors act on plastics and MPs, leading to changes in polymer properties through different degradation mechanisms [35,36,37,38,39]. Light and temperature, for example, involve free radical formation, chain scission, and subsequent reduction of molecular weight. This, together with mechanical and biotic stress, makes the polymers fragile and more susceptible to fragmentation. The formation of superficial cracks becomes, then, a site of other degradation reactions, leading to the disintegration of material [35,40].

In this context, the present study aims to assess the goodness of the most popular protocol of oxidative digestion used as a preparative step, to purify samples isolating and extracting MPs from complex environmental matrices. The method has been evaluated in terms of efficiency of extraction and recovery of MPs from the environmental matrix and, the impact and aggressiveness of the chemical digestion on the integrity of particles.

Moreover, we tested different experimental digestion conditions on virgin and aged MPs of various morphology, polymer, and size to assess if a different reaction to the chemical digestion and, an eventual alteration of items, occur based on MPs properties. We hypnotize that the rapid oxidation and the stringent exothermic reaction could destroy some polymer particles, especially the most aged ones. These particles are already fragile due to weathering caused by the time of permanence in the environment. The final objective is to advise a less impactful digestion protocol for the extraction of MPs from environmental matrices.

## 2. Materials and Methods

### 2.1. Experimental Design

Two different laboratory experiments were set up to assess the efficiency of the most used digestion protocol [14] and its impact on MPs integrity. For this purpose, the oxidative treatment, based on the Fenton reaction (Fe^2+^ + H_2_O_2_→Fe^3+^ + OH + OH^−^) [25], was tested using different temperature ranges and reagent volumes. Moreover, to reproduce the difficulties linked to the MPs extraction from complex environmental samples, virgin and aged MPs standards of different sizes and compositions were added to unpolluted soil samples. The integrity of particles and the level of alteration before and after the different treatments were evaluated through Scanning Electron Microscopy (SEM), Fourier Transformed Infrared (FTIR) spectroscopy.

In Table 1 the experimental set-up of the two trials is shown.

### 2.2. Microplastic Standards Selection

Virgin MPs of different shapes and polymers were selected by common plastic items (Table 2). Particle colour was chosen to facilitate detection and counting during the extraction phase. They were cut and smoothed in the laboratory, and particles were passed through sieves with mesh sizes from 5 mm to 1 mm, from 1 mm to 500 µm, and from 500 µm to 100 µm, obtaining MPs of three size ranges. Even, 5 mm size PE, and PP pre-production pellets were added to evaluate the impact of the most commonly used chemical digestion protocol [14] on the integrity of MPs standards (Figure 1).

### 2.3. Experiment One: Evaluating the Efficiency of MPs Digestion Treatment through Recovery Tests

#### Digestion Treatment Conditions

Virgin MPs, 30 particles for each polymer (PE, PP, PET, PVC, PS), underwent six oxidative digestion treatments at three different temperatures (75 °C, 50 °C, and 30 °C), and reagent volumes (100 or 60 mL of H_2_O_2_ + 20 mL of FeSO_4_·7H_2_O) (Table 3). The ferrous ion (Fe^2+^) of the iron sulphate heptahydrate initiates and catalyses the reaction leading to the generation of hydroxyl and hydroperoxyl radicals, powerful oxidants that degrade organic compounds [41].

Three size ranges of particles (5–1 mm; 1 mm–500 µm; 500–100 µm) were assessed for recoveries. The biggest particles (5–1 mm; 1 mm–500 µm) were added to 50 g of soil to simulate the extraction from a complex matrix while the smallest ones (500–100 µm) were added just of digestion reagents to exclude the influence of matrix on the recovery of particles and evaluate just the effect of the digestion protocol.

A solution of NaI (1.8 g cm^3^) was prepared by dissolving the salt in distilled water, to extract MPs from the environmental matrix. After the digestion treatments, the solution was added to the sample, it was shaken for about 10 s and decanted for 1 h. The supernatant was filtrated by a vacuum filtration unit (Sartorius, Goettingen, Germany) using a nitrocellulose filter (Whatman nitrocellulose membrane filters diam. 47 mm, pore size 0.45 μm) and particles were observed under a stereomicroscope (Motic SMZ – 171, Hong Kong, China).

The polymer recovery rate was calculated as the number of extracted particles on the number of added particles. The final value was expressed as a percentage.

### 2.4. Experiment Two: Evaluating the Impact of Digestion on Virgin vs. Aged MPs Integrity

#### 2.4.1. Ageing of Microplastics

Some virgin microplastics were artificially weathered in a climate room equipped with UVA lamps, calibrated at 340 nm, and programmed at a temperature of 22 °C, an irradiance of 12 h, and humidity at 60%, for a total of 20 days. Afterwards, samples were subjected to thermally ageing at 45 °C in an air-circulated oven, for another 20 days (Figure 2).

#### 2.4.2. Digestion Treatment Conditions

Three oxidative digestion treatments, with different experimental conditions, were tested on PVC, PE, PP, PS, PET fragment; PP and PE pellets, and PA fibre to evaluate the impact on virgin vs. aged MPs by SEM analysis. For each polymer, were selected particles of size from 5 mm to 500 µm and added to 13 g of soil, with the exception of PA to exclude possible contamination from fibre present in the environmental matrix (Table 4). After digestion, particles were separated by the matrix using a saturated NaI solution, and their integrity was observed by SEM, before and after treatments. Virgin and aged fibre were analyzed as tangles before treatments for the difficulty of obtaining single filaments and handling especially those being aged and fragile.

### 2.5. Fourier Transform Infrared Spectroscopy (FTIR) Acquisition

Aged particles were analyzed, before and after weathering by Fourier Transform Infrared spectroscopy using a Thermo Scientific NICOLET Summit FTIR Spectrometer (Waltham, MA, USA) equipped with an Everest ATR with a diamond Crystal plate and a DTGS KBr detector. The FTIR spectra were recorded in the region of 4000–400 cm^− 1^ with 32 scans at a resolution of 4 cm^− 1^.

### 2.6. Scanning Electron Microscopy (SEM) Acquisition

Scanning Electron Microscopy (HITACHI TM 3000 Tabletop, Tokyo, Japan) was used to observe morphology polymers before and after oxidative digestion treatments. Particles were fixed on carbon adhesive and coated with a thin layer of gold and palladium for 2 min and 10 mA to avoid charging during electron microscopy. Larger particles, such as pellets, were measured operating at 5 kV, while for other particles it was operated at 15 kV. The size of some particles was measured before and after treatments by SEM image software (Hitachi TM 3000, ver. 02-03-02, Tokyo, Japan).

### 2.7. Quality Control

A cotton coat was worn during the laboratory procedures, preventing any contamination from synthetic clothing. All glass instruments were washed three times with Milli-Q water and covered with aluminium foil. The NaI solution was filtered through a nitrocellulose filter before its use. All analytical steps were performed in a laminar flow cabinet to avoid laboratory airborne contamination.

## 3. Results

### 3.1. Results of Experiment One: Evaluating the Efficiency of MPs Digestion Treatment through Recovery Tests

As an overall result, the experimental tests performed in different temperature and peroxide volume conditions showed a recovery efficiency of about 100% for most of the MP materials used (Figure 3). The major criticalities in the extraction efficiency emerged above all for PVC and PET items.

The extraction of 1–5 mm PVC particles testing the treatment n. 1 (75 °C, 100 mL H_2_O_2_) showed the highest rate of recovery above the 100% (170 ± 1.4%) followed by treatment n. 2 (75 °C, 60 mL) with a recovery rate of 157 ± 3.5%, treatment n. 4 (50 °C, 60 mL) with 150 ± 10%, treatment n. 3 (50 °C, 100 mL) 137 ± 4%, treatment n. 5 (30 °C, 100 mL) 123 ± 4% and treatment n. 6 (30 °C, 60 mL) 122 ± 2%.

This enhancement of observed particles with respect to the initial number of added items is due to the aggressiveness of the digestion treatment on PVC, which led to its fragmentation in smaller particles observed and identified both in suspension and in the soil matrix used (Figure 4).

Likewise, the smallest PVC fragments (1 mm–500 µm; 500–100 µm) showed the same behaviour with all recoveries above 100%, confirming a fragmentation of this polymer. In each size range, the treatment n.6 (30 °C, 60 mL) resulted in having a lower impact on particles with recoveries close to 100% (116 ± 2%).

Many other small and tiny PVC particles lower than 100 µm were also observed especially in treatments 1 and 2, probably generated from the fragmentation of bigger particles. As shown in Figure 4 the surface of bigger PVC particles appears greatly modified with holes and cracks.

Differently from the PVC behaviour, all the treatments tested in the different ranges of temperature and reagents volume (treatments n. 1, 2, 3, 4, 5, and 6) demonstrated a good recovery efficiency for PS, PE and PP polymers in all the three size ranges evaluated (5–1 mm, 1 mm–500 µm and 500–100 m). Indeed, the recoveries obtained were 101 ± 1%, 100.3 ± 0.5%, 99.3 ± 0.9%, for PS, PE, and PP, respectively, showing good resistance to the oxidation reaction and an equally satisfying recovery efficiency from the matrix.

Otherwise, the recoveries of 5–1 mm and 1 mm–500 µm PET fragments showed the lowest recovery rates among all polymers, ranging from 24 and 93% of recovered particles. However, this particle loss is mostly attributable to an effect of the soil matrix used which made difficult the recovery of MPs trapping them in the bottom. Indeed, the tests on the smallest size range of PET MPs (500–100 µm), set up without the soil matrix, showed recovery rates equal to 100% for all the six treatments evaluated.

### 3.2. Results of Experiment Two: Evaluating the Impact of Digestion Treatment on Virgin and Aged MPs through Qualitative Evaluations

#### 3.2.1. Ageing of Microplastics: FTIR Acquisition

The FTIR acquisitions of MP standards of different polymer compositions made before (black lines, Figure 5) and after the ageing of particles (red lines, Figure 5), show that new absorption peaks were formed consecutively to weathering suggesting strong differences with the pristine materials probably due to their degradation (red lines, Figure 5). In the spectra of each artificially weathered particle, is evident the presence of broad peaks in the region from 3100 to 3700 cm^−1^ (OH stretching).

The ageing process produces new bands at 3423 cm^−1^ in the IR spectra of PET, PA, and PP (pellets and fragments) (Figure 5a).

In the spectra of aged PET particles, forty days after the artificial weathering, a new peak at 1614 cm^−1^, non-existent in the same particles before the ageing process, appears. (Figure 5c).

Regarding the PE particles, compared to the unaltered pristine MPs acquired by FTIR at the time zero before ageing, the weathered fragment spectra show the presence of new intense peaks at 3414 and 1577 cm^−1^ and others less intense at 873 and 777 cm^−1^. Similarly, the aged PE pellet spectra show new weathering bands at 3458 and 1618 cm^−1^.

Otherwise, the following peaks appear in the IR spectrum of PP fragments and pellets after ageing: 3404, 3440 cm^−1^, and 1643 cm^−1^ (OH bending) (Figure 5d–h). Moreover, the PP pellets show a new band at 1102 cm^−1^. The IR spectra of PS show changes corresponding to the formation of new bands at 3369 cm^−1^, 1653 cm^−1^, and 1116 cm^−1^ (Figure 5e). Regarding the analysis of the IR spectrum of weathered PVC, a broad peak of moderate intensity can be detected in the region 3000–3500 cm^−1^ and new peaks, with respect to virgin materials, are evident at 1617 cm^−1^, 1582 cm^−1^, 1193 cm^−1^, and 1148 cm^−1^ (Figure 5f).

#### 3.2.2. Scanning Electron Microscopy (SEM) Acquisition

Scanning Electron Microscopy acquisition shows the physical effects of three different oxidative treatments on the integrity of both virgin and aged particles.

Virgin MPs appear compact and solid, with a three-dimensional structure and smooth surfaces. The treatment at 30 °C (treatment c) generates a dimensional reduction of PET MPs associated with margins corrosion (Figure 6a). An expansion of the PVC (Figure 6b), showing its surface damaged by small holes, is also visible together with the PP and PS particles abrasion (Appendix A).

The treatment at 50 °C (treatment b) affects PVC and PS particles by forming holes, material loss, and corrosion (Figure 7). Even in this case, a slight reduction in the size of PET and PP fragments is visible (Appendix A).

However, the greatest surface changes occur after treatment at 75 °C (treatment a) (Appendix A). On the one hand, PVC particles manifest wide holes (Figure 8) and a lost material of PS fragment. On the other hand, PP and PET fragments show corroded margins (Figure 9). In each treatment, virgin PE and PP pellets (Figure 10) and PE fragments highlight high resistance to oxidative digestion (Appendix A).

Compared to virgin standards, aged MPs appear, before the digestion treatments, flattened, brittle, with undefined shapes with some cracks on their surface.

As well as for virgin MPs, the milder and intermediate treatments do not produce strong changes in aged particles. In all treatments, a fraying of the fibre from the initial tangle is evident.

The treatment at 30 °C generates new cracks in PVC and PS fragments, and accentuates those already present, due to weathering, in PE and PET (Appendix A).

The treatment at 50 °C produces curling of PE fragments, a fraying of the fibre, and many cracks in PET ones. Even in this case, PVC particles show a surface full of small holes (Figure 11).

The digestion treatment at 75 °C, applied on aged MPs, causes a radical alteration of most particles (Appendix A) with evident changes such as the loss of material of PVC, PS, and PP particles, cracks expansion of PET, PE corrosion, and a fraying of PA (Figure 12). Aged pellets show several abrasions on their surface, especially after the most aggressive treatment at 75 °C (Figure 13).

## 4. Discussion

Chemical digestion treatment is a crucial step of MPs analysis, especially when environmental samples are rich in organic matter; indeed, removing the natural debris promotes the subsequent extraction of MPs from the matrix. However, the choice of the protocol must consider its digestion effectiveness and, at the same time, the integrity of polymers, so as not to compromise the identification and quantification of particles.

In this regard in the present study, attention has been paid to the impact that one of the most common digestion protocols, based on Fenton’s reaction, could have on MPs in terms of recoveries and integrity of particles.

In experiment number one, we hypothesized that the recovery of virgin MPs may be subjected to the type of sample in which the particles are dispersed. Although the density of the NaI solution, adopted for MPs extraction, is higher than high-density polymer standards (e.g., PVC and PET), it was not always possible to fully recover all particles.

Most low-density polymers were recovered almost always at 100%, instead the PET of dimensions from 5 mm to 500 microns, and the PVC of dimensions between 1 mm and 500 microns, were recovered with low efficiency because they remained inside the matrix. A similar result was also observed in a previous study [42], where small PVC fragments were trapped in the sediment after the density separation, but the protocol used does not correspond to this study. Nevertheless, the recoveries of these two polymers were also demonstrated to be low in other studies [43,44]. This behaviour may depend on the effect of the oxidation of the polymer surface, which leads to increased hydrophilicity, reducing the number of possible air bubbles and the buoyancy of the material [43,45]. Previous studies have already reported a relationship between size and recovery rate highlighting that separation of particles with diameters lower than one mm is more difficult than large MPs [27,46,47]. Electrostatic interactions between MPs and other particles depend on the morphology, composition, size, and surface charge of MPs [48]. Therefore, the presence of the matrix can interfere with the MPs extraction phase from the soil and our study suggests this with the recovery of 100% of PET, in the absence of soil.

Our results also highlight the presence of more PVC particles in the suspension and in the soil than initially added. It is appended both in the presence of the matrix and in the absence, after all treatments, especially at 75 °C using 100 mL of H_2_O_2_, suggesting the influence of temperature in polymer fragmentation. Indeed, an increase in the temperature of the digestion treatment caused the loss of some types of MPs also in previous studies [27,28]. The temperature is an important factor in the extraction procedure as it can affect the characteristics of a polymer based on its glass transition temperature (Tg), which is the critical temperature at which a material changes its features from “glassy material” to “rubbery material” [49].

In experiment number two, we obtained a visual demonstration of how Fenton’s reagent impacts MPs. In this case, both virgin and aged particles were analyzed with the addition of PE and PP pellets, and PA fibres. Various previous studies tested digestion protocols on polymers but pellets are the most used type followed by fragments and fibres [27,28,29,50,51].

The greater number of small PVC particles counted could be explained by observing the formation of large pores with a slight expansion of the size of the PVC polymer. These pores, favoured by an increase in temperature, may generate a release of tiny fragments of PVC such as those found in the soil. The presence of larger cavities has been reported in [52] for some polymers after the Fenton oxidation process.

SEM analysis also shows a size reduction and margin corrosion of PET and PP, after treatment. Even in [53] particles of PE and PP show a size reduction of about 10%, after long treatment with H_2_O_2_ but in our case, virgin PE showed high resistance to treatment along with pellets. The corrosion of PET margins and the formation of roughness may have favoured the creation of active sites on the polymer increasing the adhesion of soil particles on the MPs surface. In addition, a particle size reduction could affect those results that include the evaluation of MPs size ranges because particles of a lower size range may not be detected because they are strongly aggravated by the treatment.

As for the virgin PS, particles were damaged at 50 °C and 75 °C as reported by [27]. We have not observed any major changes in this polymer despite the known damage caused by a temperature from 70 °C to 100 °C on the PA [29,54]. However, our data cannot be fully compared with these studies because we show a morphological alteration of the particles and not an assessment of weight.

The results observed so far suggest that the oxidative treatment generates an impact on virgin particles caused by high temperatures. However, environmental MPs are particles already altered and damaged due to biotic and abiotic degradation [35,55,56].

Indeed, the FTIR acquisitions highlighted the formation of new peaks in the polymer spectra consecutively to the ageing of virgin polymers in artificial conditions. As previously observed by other authors [57,58,59], the regions reflecting ageing-related changes (hydroxyl groups, peaks from 3100 to 3700 cm^−1^, carbon double bonds, 1600 and 1680 cm^−1^ and carbonyl groups, 1690 and 1810 cm^−1^) appeared greatly modified compared to the pristine materials in each type of MPs.

Several previous works conducted under simulated environmental conditions and on field-collected samples showed that photo and thermal oxidation together with humidity could alter the physicochemical structure of MPs leading to the introduction of oxygen into the polymer chain with the formation of carbonyl (CO) and hydroxyl (OH) functional groups [60,61,62,63].

As a matter of fact, in the spectra of our degraded MPs, exposed to UVA followed by a period of incubation at 45 °C in dry conditions, new broad hydroxyl peaks (centred at 3300–3400 cm^− 1^), appeared in the aged particles and were the most readily identified. The forced weathering, resulted, in the concomitant occurrence of oxidation reactions, chain degradation, and the formation of surface cracks and fractures. This alteration could allow an easier and deeper infiltration of water and oxygen from the atmosphere into the sample leading, in time, to an increased effect of ageing [64]. However, our weathered samples have been dried immediately before FTIR acquisition, therefore, the OH bands origin could be minimally related to the presence of adsorbed humidity from the atmosphere and more likely to the polymeric alcohols, used as a lubricant in plastic, or other by-products easily released during the degradation process [5].

Moreover, the entire region between 1550 and 1810 cm^−1^ usually referred to as the “carbonyl groups” is indicative of oxidized carbon in the plastic hydrocarbon chain. The presence of carbonyls, in almost all our weathered polymers, even if less distinctive compared to hydroxyl groups, suggests that oxygen has bonded with the hydrocarbon chain [60,65,66].

Indeed, the use of a “carbonyl index” (CI) is frequently used to measure the light-induced photo-oxidation, since it increases with increasing exposure time of plastic into the environment or progressive ageing of MPs [57,59,67].

Moreover, in our experiments, new peaks are present almost all in the fingerprint region (1500–500 cm^−1^).

However, the spectral changes during these processes are not yet fully understood and it is complex to monitor and predict these changes.

The increased peaks around 1760 and 1690 in PVC, PP, and PE fragment spectra, indicates the presence of carbonyl groups formed by photo-oxidation in the climate chamber [57,67].

Therefore, the assessment of methodological protocols for MPs studies should consider the intrinsic physicochemical difference between virgin and aged starting materials preferring the use of weathered particles to simulate the real environmental conditions. To our knowledge, only one study used aged particles for protocol testing [18].

Compared to virgin particles, PA fibres and PE fragments, already damaged by the ageing process, showed signs of breakage and fraying. The impact of the treatment temperature was strongly evident on the aged particles because it caused the loss of polymeric material and a strong alteration of all particles. Even aged pellets showed a rough surface already after the milder treatments until to show the most evident abrasions after the digestion at 75 °C.

## 5. Conclusions

This study evaluated the impact of the most common oxidative digestion protocol used to extract MPs from environmental samples. We have provided a visual demonstration of particle alterations by SEM analysis and we find that this technique can help in the evaluation of protocols. Wet peroxide oxidation is an effective organic digestion method for different environmental matrices but our results showed morphological changes in polymers not observed in other studies. Both virgin and aged MPs were damaged, especially after treatment at 75 °C. Several factors must be considered in the assessment of experimental conditions. Furthermore, methods recommending temperatures below 50 °C should be preferred at the expense of longer digestion times. We also suggest evaluating the type of matrix, particle size, and shape and exploring a broad type of polymer in methodological analysis, to ensure a comprehensive assessment of the impact on MPs. Moreover, the weathering of particles in simulated environmental conditions, showed great alterations of FTIR spectra influencing the correct polymer characterization of real environmental samples and making difficult the interpretation of spectra. The degree of degradation is probably connected to the time of exposure of particles to environmental weather conditions.

Nonetheless, the evaluation of methodological approaches on aged MPs rather than on pristine materials is essential to ensure a more realistic vision of obtained results and a better quality of the final data. 

## Figures and Tables

**Figure 1 polymers-14-01958-f001:**
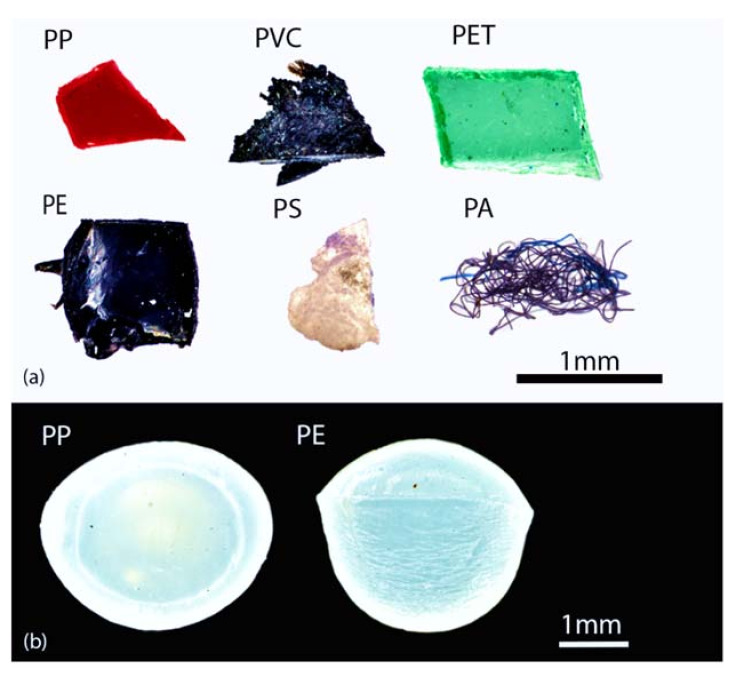
(**a**) Virgin MPs products by cutting common plastic items. (**b**) Pre-production pellets added in experiment two. Images produced by Carl Zeiss Tessovar Microscope.

**Figure 2 polymers-14-01958-f002:**
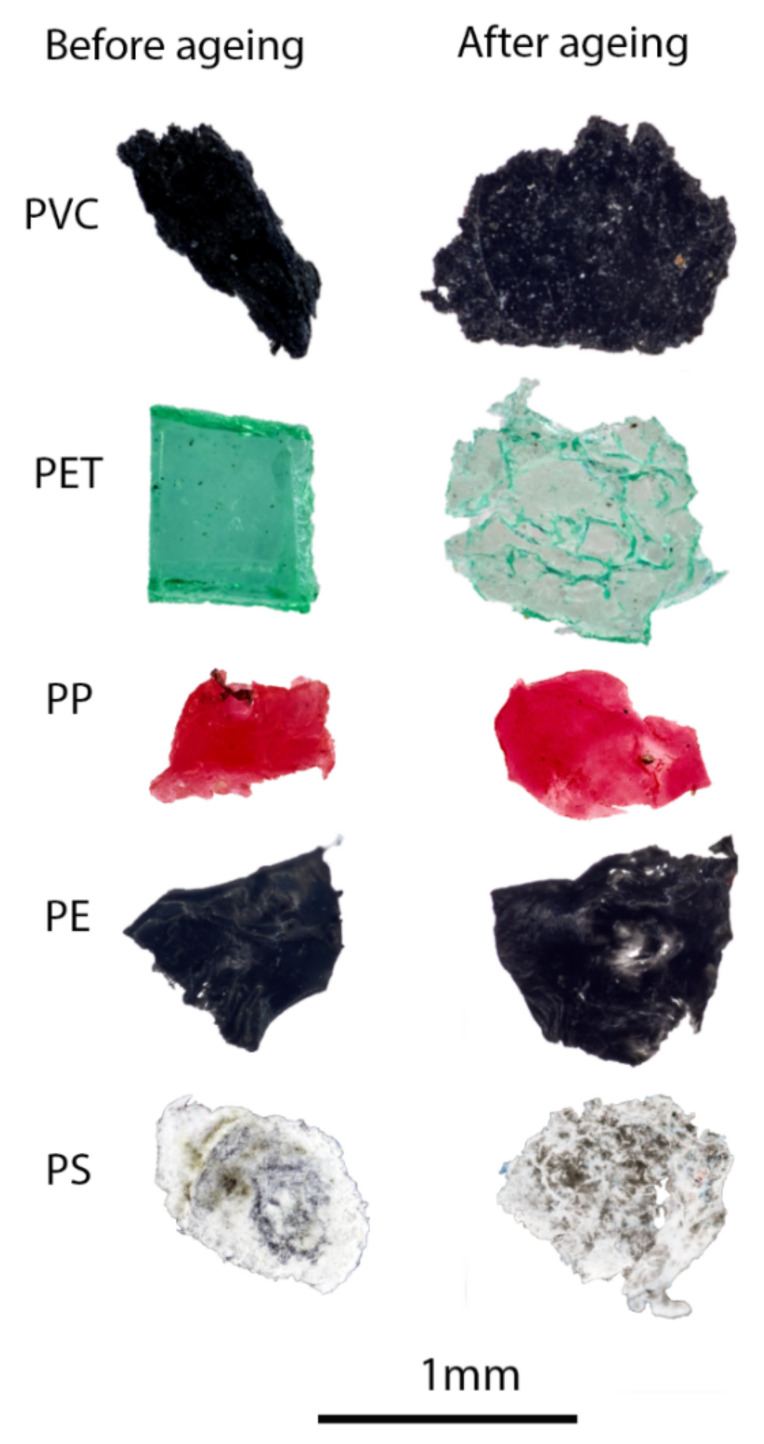
The morphological aspect of some polymers (PVC, PET, PP, PE, PS) before and after the ageing process. These polymers, together with PA fibre and pellets (PP, PE), were exposed to UVA (photo-oxidation) in the climatic chamber for 20 days and then at a temperature of 45 °C for a further 20 days in dry conditions.

**Figure 3 polymers-14-01958-f003:**
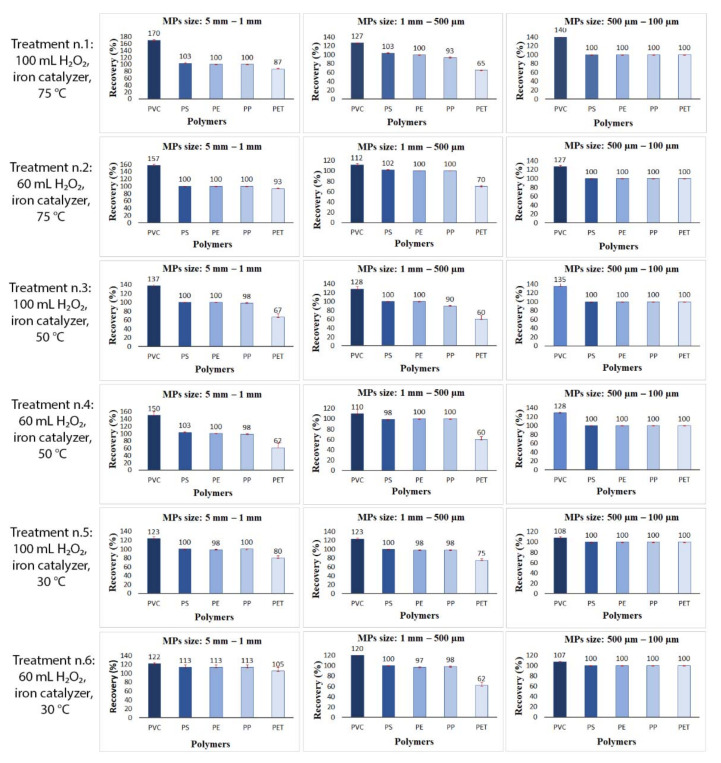
Recovery rates of PVC, PS, PE, PP, and PET, of three-dimensional sizes, after six different treatments varying for H_2_O_2_ volumes and temperatures. Values are expressed as a percentage mean value of two replicates ± the standard deviation.

**Figure 4 polymers-14-01958-f004:**
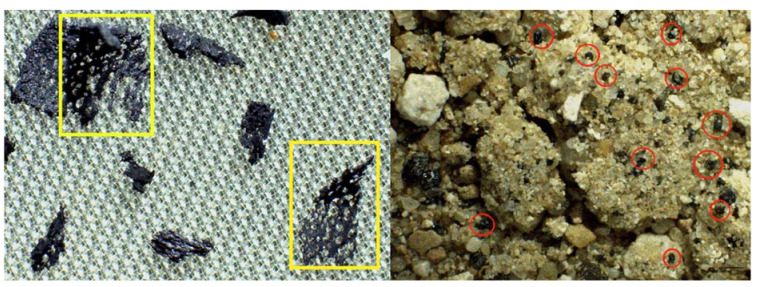
Particulars of PVC fractures (yellow boxes on the left) due to the abrupt oxidation reaction at 75 °C that led to the fragmentation of PVC in tiny particles trapped in the soil matrix (red circles on the right).

**Figure 5 polymers-14-01958-f005:**
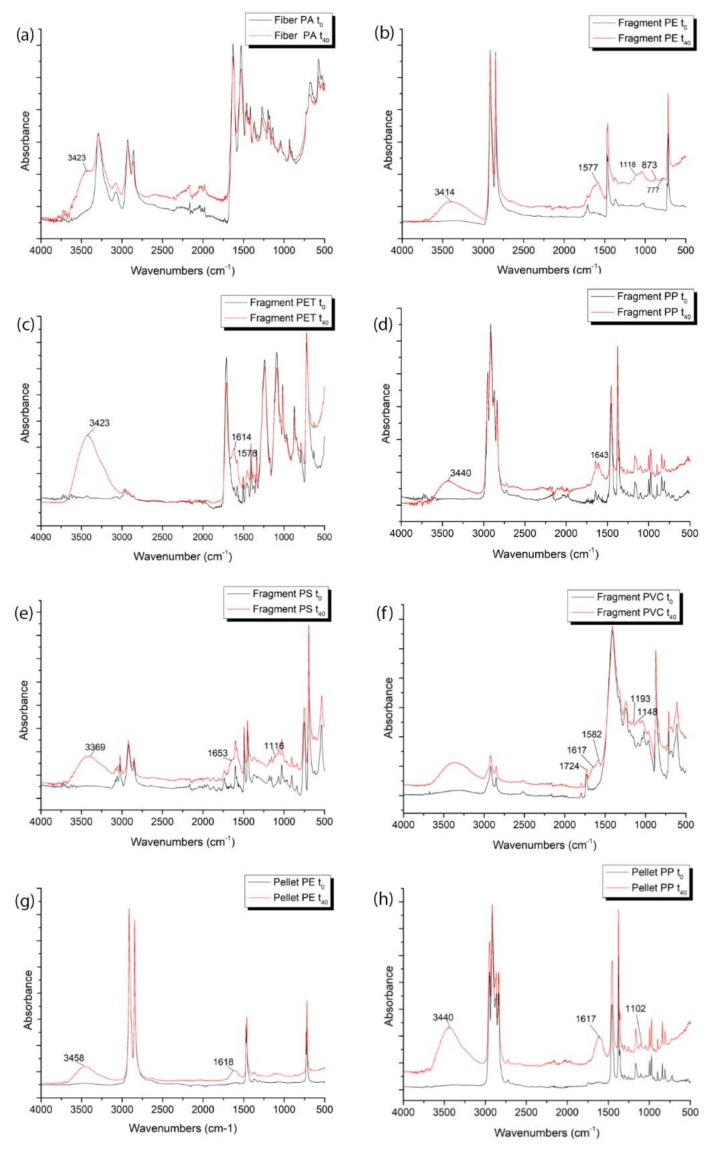
Comparison between the IR spectra of different polymers before (black lines, t_0_) and after 40 days of ageing (red lines, t_40_). New peaks formed after ageing are indicated in the spectra of each polymer. Absorption areas related to ageing from 3100 to 3700 cm^−1^ (hydroxyl groups) are evident in all polymers. (**a**) PA fiber; (**b**) PE fragment (**c**) PET fragment; (**d**) PP fragment; (**e**) PS fragment; (**f**) PVC fragment; (**g**) PE pellet; (**h**) PP pellet.

**Figure 6 polymers-14-01958-f006:**
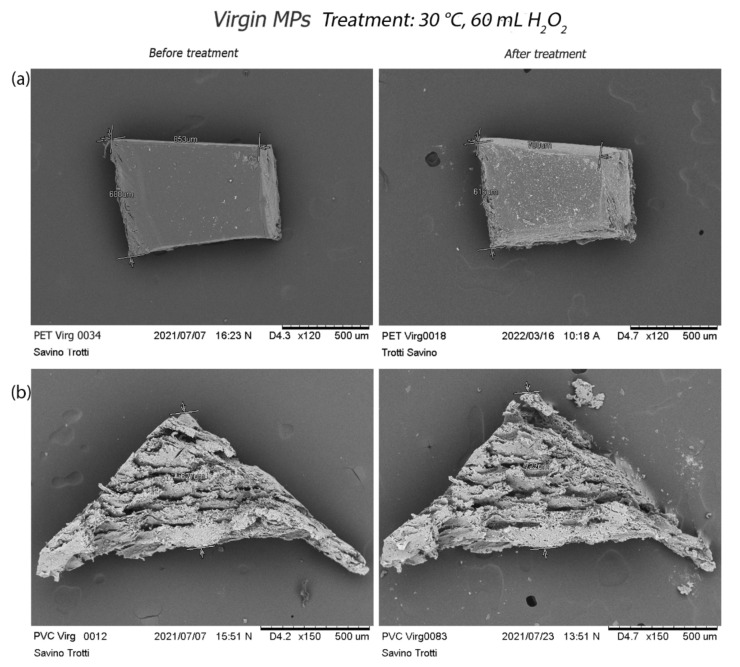
Alterations caused by treatment at 30 °C on virgin PET and PVC: (**a**) size reduction and corrosion of virgin PET margins, (from 853 to 708 µm); (**b**) PVC dimensional expansion from 627 µm to 722 µm.

**Figure 7 polymers-14-01958-f007:**
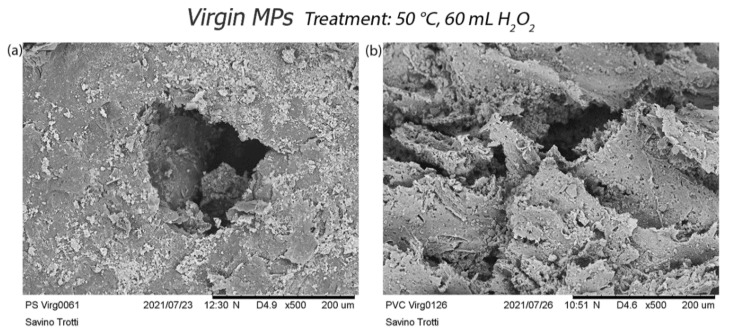
Effects of treatment at 50 °C on virgin PS and PVC: (**a**) formation of a hole on the surface of the PS; (**b**) PVC particle surface with small holes.

**Figure 8 polymers-14-01958-f008:**
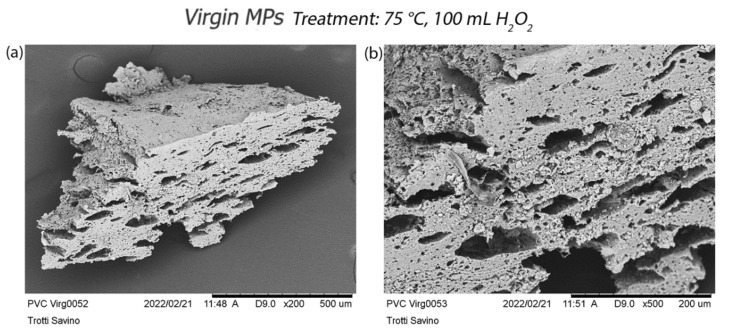
Effects of oxidative digestion treatment at 75 °C on the virgin PVC: (**a**) morphological acquisition of the entire PVC particle after treatment; (**b**) detail of large holes inside the PVC particle.

**Figure 9 polymers-14-01958-f009:**
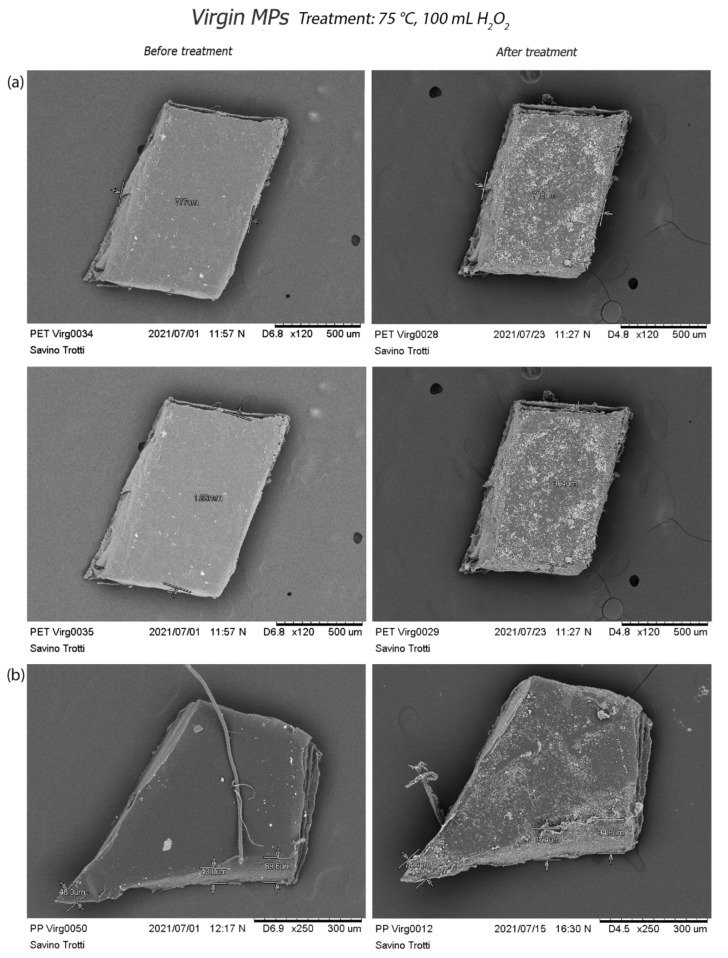
Corrosive treatment effects on PET and PP particles: comparison of size measurements, before and after treatment, emphasizes the corrosion of virgin PET (**a**) and PP (**b**).

**Figure 10 polymers-14-01958-f010:**
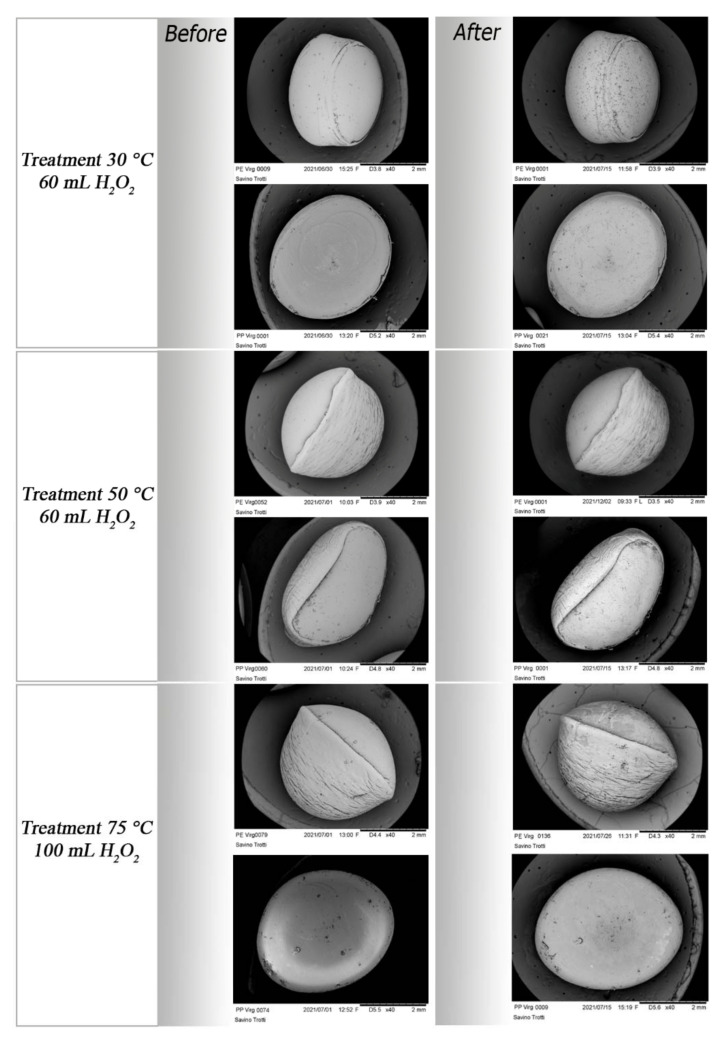
Morphological aspects of pellets before and after different treatments. Virgin PE and PP pellets highlight high resistance to oxidative digestion.

**Figure 11 polymers-14-01958-f011:**
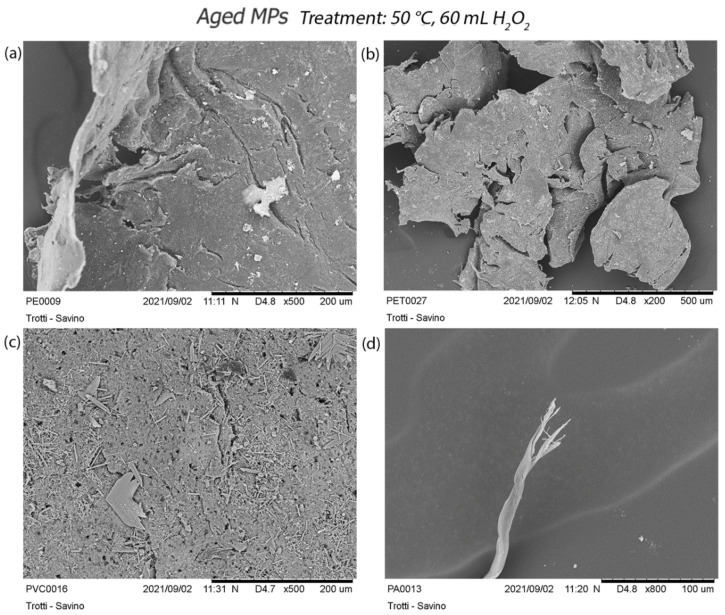
Focus on the effects of treatment at 50 °C on aged particles: (**a**) detail of PE curling; (**b**) more cracks in the PET; (**c**) small holes on the PVC surface; (**d**) fraying of PA fibre.

**Figure 12 polymers-14-01958-f012:**
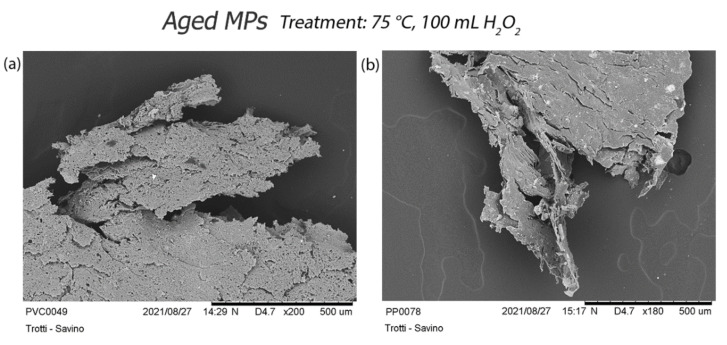
Focus on the effects of treatment at 75 °C on aged particles: (**a**) small holes on the PVC surface and loss of the polymer material; (**b**) corrosion and loss of PP polymer material; (**c**) formation of large cracks in PET; (**d**) corrosion and loss of PE polymer material; (**e**) breaking and fraying of the fibre.

**Figure 13 polymers-14-01958-f013:**
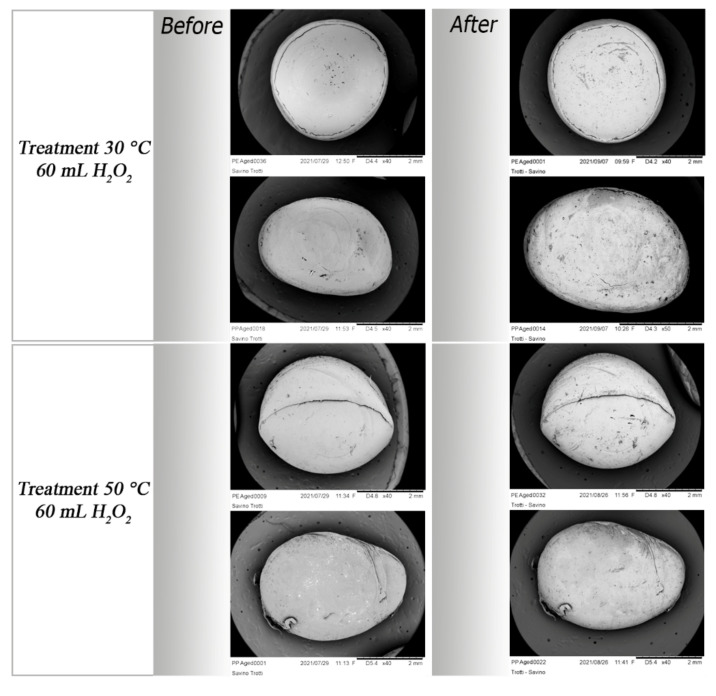
Morphological aspects of aged pellets before and after different treatments: abraded areas are highlighted as the temperature increases.

**Table 1 polymers-14-01958-t001:** Indication of the experimental conditions followed for each trial.

Experiment One	
AIM	Evaluate the efficiency of extraction of the most commonly used chemical digestion protocol (based on Wet Peroxide Oxidation [14]) on the recovery of virgin MPs standards from a complex matrix
Particle selection	Virgin MPs
Matrix selected	Soil
Starting digestion condition	Reagents volume: 20 mL of 30% H_2_O_2_ solution add to 20 mL of 0.05 M iron sulphate heptahydrate (FeSO_4_·7H_2_O) every 30′ until complete sample digestion. The temperature of reaction: 75 °C.
Density separation	NaI (1.8 g cm^3^)
Qualitative analysis	Stereomicroscope
**Experiment Two**	
AIM	Evaluate the impact of the most commonly used chemical digestion protocol on the integrity of virgin and aged MPs standards
Particle selection	Virgin and aged MPs
Matrix selected	Soil
Starting digestion condition	Reagents volume: 20 mL of 30% H_2_O_2_ solution add to 20 mL of 0.05 M iron sulphate heptahydrate (FeSO_4_·7H_2_O) every 30′ until complete sample digestion. The temperature of reaction: 75 °C.
Density separation	NaI (1.8 g cm^3^)
Qualitative analysis	FTIR—SEM

**Table 2 polymers-14-01958-t002:** Polymer, density, source, colour, and shape of MPs selected as standards for the experiments. (*): image reworked from source [10].

Polymers	Density (g cm ^3^) (*)	Source	Colour	Shape
Polystyrene (PS)	0.01–1.06	Food box	White	Fragment
Polypropylene (PP)	0.85–0.92	Disposable glass	Red	Fragment
Polyethylene (PE)	0.89–0.98	Mulching films	Black	Fragment
Polyamide (PA)	1.12–1.15	Textile	Black	Fibre
Polyvinyl chloride (PVC)	1.38–1.41	Building material	Black	Fragment
Polyethylene terephthalate (PET)	1.38–1.41	Plastics bottle	Green	Fragment

**Table 3 polymers-14-01958-t003:** Summary of the different oxidative digestion conditions used in experiment one. “-” Treatment made in the absence of the matrix on particle sizes from 500 to 100 µm.

Treatment	Reagent Volumes	Temperature (°C)	Polymers	Size	Soil Matrix (g)
**1**	100 mL H_2_O_2_ + 20 mL FeSO_4_·7H_2_O	75 °C	PE, PP, PET, PVC, PS	5–1 mm 1 mm–500 µm	50
				500–100 µm	-
**2**	60 mL H_2_O_2_ + 20 mL FeSO_4_·7H_2_O	75 °C	PE, PP, PET, PVC, PS	5–1 mm 1 mm–500 µm	50
				500–100 µm	-
**3**	100 mL H_2_O_2_ + 20 mL FeSO_4_·7H_2_O	50 °C	PE, PP, PET, PVC, PS	5–1 mm 1 mm–500 µm	50
				500–100 µm	-
**4**	60 mL H_2_O_2_ + 20 mL FeSO_4_·7H_2_O	50 °C	PE, PP, PET, PVC, PS	5–1 mm 1 mm–500 µm	50
				500–100 µm	-
**5**	100 mL H_2_O_2_ + 20 mL FeSO_4_·7H_2_O	30 °C	PE, PP, PET, PVC; PS	5–1 mm 1 mm–500 µm	50
				500–100 µm	-
**6**	60 mL H_2_O_2_ + 20 mL FeSO_4_·7H_2_O	30 °C	PE, PP, PET, PVC, PS	5–1 mm 1 mm–500 µm	50
				500–100 µm	-

**Table 4 polymers-14-01958-t004:** Summary of different oxidative digestion conditions used in experiment two. “-” Treatment made in the absence of the matrix on particle sizes from 5 to 1 mm.

Treatment	Reagent Volumes	Temperature (°C)	Polymers	Size	Soil Matrix (g)
**a**	100 mL H_2_O_2_ + 20 mL FeSO_4_·7H_2_O	75 °C	PE, PP, PET, PVC, PS	5–1 mm 1 mm–500 µm	13
			PA	5–1 mm	-
**b**	60 mL H_2_O_2_ + 20 mL FeSO_4_·7H_2_O	50 °C	PE, PP, PET, PVC, PS	5–1 mm 1 mm–500 µm	13
			PA	5–1 mm	-
**c**	60 mL H_2_O_2_ + 20 mL FeSO_4_·7H_2_O	30 °C	PE, PP, PET, PVC, PS	5–1 mm 1 mm–500 µm	13
			PA	5–1 mm	-

## Data Availability

Not applicable.

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
