# Peer review of "Effects and Impacts of Different Oxidative Digestion Treatments on Virgin and Aged Microplastic Particles"

_polymers, 2022, doi:10.3390/polym14101958_

Round 1
Reviewer 1 Report
The paper by Claudia Campanale et al deals with aging of microplastic particles. They used Fenton reaction at different conditions in order to simulate the aging.
Main points:
- Readability of the manuscript is pure. It contain many large figures supplemented with short text parts, which does not help much. For example, Section 3.1 consists of half page of text and three almost one-page figures.
- Analysis and discussion of IR data is not correct. From spectra presented, I only see water absorption evident from appearance of new OH stretching (~3400 cm-1), OH bending (~1600 cm-1) bands. Suggested appearance of other “functions” is hard to follow.
- Appearance of CC double bond, suggested by IR analysis, is hardly acceptable from chemical point of view.
- Scientific outcome of the study is not clear. Is Fenton reaction able to destroy at least some types of microplactic particles? Could it help to remove them from environment?
- Here described experiments could be taken as an accelerated model of aging in environment. For this purpose, however, control samples from degradation in environment are necessary.
In summary, I cannot recommend this article for publication in current form.
Author Response
Reviewer 1
The paper by Claudia Campanale et al deals with aging of microplastic particles. They used Fenton reaction at different conditions in order to simulate the aging.
Dear reviewer, thank you for your comments and suggestions that help us to enhance our manuscript quality. We thoroughly revised the work increasing references, improving image quality and quantity, data elaboration, discussions and the general aim of the work. We agree with you with many of your comments and suggestions and we tried to improve the work by taking into account your comments. Changes in the revised version of the manuscript are evidenced in blue.
Main points:
- Readability of the manuscript is pure. It contain many large figures supplemented with short text parts, which does not help much. For example, Section 3.1 consists of half page of text and three almost one-page figures.
Thank you for your comment. We worked on this aspect. We reduced and condensed the figures of section 3.1 in just one figure, now figure 3. We also improved data adding means of recoveries and standard deviations and improving the text with the results description. Please see lines 217-227.
- Analysis and discussion of IR data is not correct. From spectra presented, I only see water absorption evident from appearance of new OH stretching (~3400 cm-1), OH bending (~1600 cm-1) bands. Suggested appearance of other “functions” is hard to follow.
Thank you for your comment. We modified the IR description in the results section (please see section 3.2.1) and improved its discussion (see lines 462-471). However, as demonstrated and cited by other authors that we cited in the discussion, the formation of new peaks in the aged polymers (aged in simulated environmental conditions) compared to pristine materials has been already showed to be related to weathering-related changes such as the photooxidation. See also the reference of Gonzalez et al., 2021.
- Appearance of CC double bond, suggested by IR analysis, is hardly acceptable from chemical point of view.
As the previous point, we modified this section and its discussion.
- Scientific outcome of the study is not clear. Is Fenton reaction able to destroy at least some types of microplactic particles? Could it help to remove them from environment?
Dear reviewer, thank you for your comment. Maybe we didn’t give a sufficient description of the aim of the work and we didn’t explain it well. The aim of our work is: “ the present study aims to assess the goodness impact and efficiency of the most popular protocol of oxidative digestion used as a preparative step, to purify samples isolating and extracting microplastics from complex environmental matrices. The method has been evaluated in terms of efficiency of extraction and recovery of microplastics from the environmental matrix and, impact and aggressiveness of the chemical digestion on the integrity of particles. Moreover, we tested different experimental digestion conditions on virgin and aged MPs of various morphology, polymer, and size to assess if a different reaction to the chemical digestion and, an eventual alteration of items, occur based on microplastic properties. We hypnotize that the rapid oxidation and the stringent exothermic reaction could destroy some polymer particles, especially the most aged ones. These particles are already fragile due to weathering caused by the time of permanence in the environment. The final objective is to advise a less impactful digestion protocol for the extraction of microplastics from environmental matrices. “
We modified and improved the main purpose of our work and reported at lines 75-90.
- Here described experiments could be taken as an accelerated model of aging in environment. For this purpose, however, control samples from degradation in environment are necessary.
We didn’t understand your question. Our experiments are not an accelerated model of ageing in the environment. Our experiments simulate a pre-treatment, a purification step NECESSARY to purify complex environmental samples (e.g. soils, wastewater but also surface waters) aimed to extract and isolate microplastics from the matrix and quantify and characterize them. About all researchers that work on microplastics adopt these kinds of protocols for microplastics analyses. The oxidative digestion (Fenton’s reaction) is needed to destroy the natural organic fraction present in soil or waters (e.g. leaves, little organisms, natural debris, etc.).
In summary, I cannot recommend this article for publication in current form.

Reviewer 2 Report
Recommendation: Minor revisions needed.
Comments:
The paper by Savino et al. contributes the aspect of evaluate the integrity of MPs of different polymeric compositions, sizes, and morphology after Fenton’s reaction digestion. The title and abstract are appropriate for the content of the text. Moderate English changes are required. The article gives an interesting historical and scientific perspective on methods for analyzing microplastics (MPs).
Some issues should be addressed prior to publication.
- “analysing” should be spelled as analyzing.
- There are not enough references to the studies that analyzed the effects of digestion treatments on MPs. Please include the most recent and important one. “In the environment, biotic and abiotic factors act on plastics and MPs, leading to changes in polymer properties through different degradation mechanisms.” You should include the reference for enzymatic degradation mechanism of polymer such Enzymatic polymerization of poly (glycerol-1, 8-octanediol-sebacate): Versatile poly (glycerol sebacate) analogues that form monocomponent biodegradable fiber scaffolds. Biomacromolecules, 21(8), 3197-3206.
- Table 1. Please regulate the format of the contents.
- Table 2. Since the data here is not from your own characterization, please add a reference.
- Table 3. What do you mean by 7H2O2?
- Figure 2. Why do you select this few pieces of polymers after aging test? What are the degradation conditions for each of them?
- Figure 3, Figure 4, and Figure 5. The 3D column is not necessary here. Instead, you should add the data of recovery of each polymer on the top of their columns.
- Figure 6. Please label the critical peak on the graphs, which peak are we looking at here? Which peak shows dramatic change through the time, you should give that information on your graph.
In summary, it can be seen that the author has done a lot of work on the manuscript. But there are still some deficiencies that need to be improved, especially for the explanation of graphs and phenomena. The experimental scheme also needs to be further enhanced.
Author Response
Reviewer 2
The paper by Savino et al. contributes the aspect of evaluate the integrity of MPs of different polymeric compositions, sizes, and morphology after Fenton’s reaction digestion. The title and abstract are appropriate for the content of the text. Moderate English changes are required. The article gives an interesting historical and scientific perspective on methods for analyzing microplastics (MPs).
Dear reviewer, thank you for your comments and suggestions that help us to enhance our manuscript quality. We thoroughly revised the work increasing references, improving image quality and quantity, data elaboration, discussions and the general aim of the work. We agree with you with your comments and suggestions and we tried to improve the work by taking into account your comments. Changes in the revised version of the manuscript are evidenced in blue.
Some issues should be addressed prior to publication.
analysing” should be spelled as analyzing.
We changed the spelling of “analysing” in “analyzing”;
There are not enough references to the studies that analyzed the effects of digestion treatments on MPs. Please include the most recent and important one. “In the environment, biotic and abiotic factors act on plastics and MPs, leading to changes in polymer properties through different degradation mechanisms.” You should include the reference for enzymatic degradation mechanism of polymer such Enzymatic polymerization of poly (glycerol-1, 8-octanediol-sebacate): Versatile poly (glycerol sebacate) analogues that form monocomponent biodegradable fiber scaffolds. Biomacromolecules, 21(8), 3197-3206.
lines 61-64: we added more references about digestion treatments;
lines 61-69: we added the most recent and important references about studies that analyzed the effects of digestion treatments on MPs; we also added the reference that you suggested
The new references added are: 25, 28, 19, 27, 29-33, 34, 35-39
Table 1. Please regulate the format of the contents.
we reformatted the table but probably it is altered in the pdf version;
Table 2. Since the data here is not from your own characterization, please add a reference.
we added a reference about the density of polymers because the other information is our production. We chose the color, polymers and type of plastic material from which to produce the particles;
Table 3. What do you mean by 7H2O2?
thank you for the observation because we realized the error in writing the iron sulfate heptahydrate. Therefore, we corrected it and inserted in lines 126 -129 more information about the role of this reagent as catalyser of reaction.
Figure 2. Why do you select this few pieces of polymers after aging test? What are the degradation conditions for each of them?
in the initial figure, only some of the polymers subjected to the aging process have been chosen as an example. However, we integrated with the PE and PS particles and added information about the degradation conditions.
Figure 3, Figure 4, and Figure 5. The 3D column is not necessary here. Instead, you should add the data of recovery of each polymer on the top of their columns.
by collecting the suggestions of all the reviewers, we reduced to a single figure the graphical representation of the recovery rates, so as not to have three large images and occupy too much space. We deleted the 3D column and added both data of recovery of each polymer on the top of their columns and standard deviation.
Figure 6. Please label the critical peak on the graphs, which peak are we looking at here? Which peak shows dramatic change through the time, you should give that information on your graph.
we modified the figure by adding the critical peak and some information about changes in IR spectra.
In summary, it can be seen that the author has done a lot of work on the manuscript. But there are still some deficiencies that need to be improved, especially for the explanation of graphs and phenomena. The experimental scheme also needs to be further enhanced.

Reviewer 3 Report
I have the following comments:
- The aim of the research should be clearly stated in the abstract.
- Line 58: the reference is missing.
- Line 72: this sentence is not in the proper place. Authors should not describe in this section, at the end of the Introduction section only the aim of the study should be explained.
- The introduction part should contain at least one part about the alternatives of plastic. The following reference can be used: Dordevic, D., Necasova, L., Antonic, B., Jancikova, S. and Tremlová, B., 2021. Plastic cutlery alternative: Case study with biodegradable spoons. Foods, 10(7), p.1612.
- The statistical analysis is not included, why?
- Figure 3: statistical analysis is not included, but also authors did not include standard deviation in the figure too.
- Figure 4: statistical analysis is not included, but also authors did not include standard deviation in the figure too.
- Figure 5: statistical analysis is not included, but also authors did not include standard deviation in the figure too.
Author Response
Reviewer 3
Dear reviewer, thank you for your comments and suggestions that help us to enhance our manuscript quality. We thoroughly revised the work increasing references, improving image quality and quantity, data elaboration, discussions and the general aim of the work. We agree with you with your comments and suggestions and we tried to improve the work by taking into account your advices. Changes in the revised version of the manuscript are evidenced in blue.
- The aim of the research should be clearly stated in the abstract.
Thank you for your comment. We better focus the aim of the work both in the abstract and in the Introduction. Please see lines 16-19 (abstract) and 75-90 (introduction)
- Line 58: the reference is missing.
We added the reference “Munno, K.; Helm, P.A.; Jackson, D.A.; Rochman, C.; Sims, A. Impacts of temperature and selected chemical digestion methods on microplastic particles. Environ. Toxicol. Chem. 2018, 37, 91–98, doi:10.1002/etc.3935.”
- Line 72: this sentence is not in the proper place. Authors should not describe in this section, at the end of the Introduction section only the aim of the study should be explained.
Ok, thank you. We moved the sentence in the Materials & Methods. See lines 99-101
- The introduction part should contain at least one part about the alternatives of plastic. The following reference can be used: Dordevic, D., Necasova, L., Antonic, B., Jancikova, S. and Tremlová, B., 2021. Plastic cutlery alternative: Case study with biodegradable spoons. Foods, 10(7), p.1612.
Thank you for the observation to include a part about alternative plastics in the introduction. We tried to link it to the context, although we find it too different from the purpose of the work, we hope we succeeded in the best. We added the reference that you suggested (34)
- The statistical analysis is not included, why?
- Figure 3: statistical analysis is not included, but also authors did not include standard deviation in the figure too.
- Figure 4: statistical analysis is not included, but also authors did not include standard deviation in the figure too.
- Figure 5: statistical analysis is not included, but also authors did not include standard deviation in the figure too.
Thank you for your observations. We provide replicates for each trial and we calculated mean values and standard deviations as you suggested. We modified the graphs by adding the average percentage value of recovery rates for each polymer, plus the standard deviation. We included in the text this information, also.

Round 2
Reviewer 1 Report
The paper by Claudia Campanale et al has been improved considerably but the issue related with infrared spectra interpretation remains.
- In the revised paper, authors removed all assignments of the IR bands, which avoids to errors but does not help the interpretation of the spectra. In my opinion, the assignment of the bands at ~3100-3700 cm-1 (OH stretching) and ~1620-1650 cm-1 (OH bending) should remain in the paper, as it is unambiguous.
- Line 277: I cannot agree with statement: “The FTIR acquisitions of MP standards … demonstrate some substantial changes in the chemical structure of polymers…” since origin of OH bands is not clear. Of course, the polymers can be functionalized by OH groups but there is an alternative explanation not covering chemical changes. The changes in polymer morphology, observed by other experiments, can improve absorption power of the material and appeared OH bands are due to absorbed water.
Author Response
Dear reviewer, thank you for your further revision that helped us to increase the manuscript quality more.
We agree with you that the interpretation of FTIR spectra of our weathered particles, exposed to simulated environmental conditions, is not so easy.
We revised the statement on line 277. Please see now lines 273-276. We added OH bending e OH stretching as you suggested. See lines 277 and 289.
We tried to give an interpretation of spectra and new peaks of degraded particles in the discussion section. Please see the integration on lines 484-512. We agree that the change in polymer morphology could help an easier infiltration of atmospheric humidity. However, maybe we didn’t specify, that our ageing trials have been conducted in dried conditions and spectra acquired immediately after a period of incubation at 45°C. This cannot absolutely exclude the presence of water explaining the OH bands but maybe it is minimal. However, the FTIR acquisitions of weathered particles demonstrated that particles changed a lot (from a physical and maybe also from a chemical point of view) after ageing. This could be a problem for the polymer identification of unknown particles found in the environment due to the great alteration of spectra.
We specify all this in the discussion written in blue.
Moreover, we implemented also the conclusions based on the new FTIR discussion.
Reviewer 3 Report
The manuscript can be accepted.
Author Response
Thank you for your revision
Round 3
Reviewer 1 Report
OK